# The Evolving Landscape: Exploring the Future of Myelodysplastic Syndrome Treatment with Dr. Rami Komrokji

**DOI:** 10.3390/cancers15215170

**Published:** 2023-10-27

**Authors:** Sean Henry Jackewicz, Helena S. Coloma, Viviana Cortiana, Muskan Joshi, Gayathri P. Menon, Maduri Balasubramanian, Chandler H. Park, Yan Leyfman

**Affiliations:** 1School of Medicine, Mercer University, Savannah, GA 31404, USA; 2Harvard University, Cambridge, MA 02138, USA; 3Department of Medical and Surgical Sciences (DIMEC), University of Bologna, 40126 Bologna, Italy; 4Tbilisi State Medical University, Tbilisi 0186, Georgia; 5Norton Cancer Institute, Louisville, KY 40241, USA; 6Icahn School of Medicine at Mount Sinai South Nassau, Oceanside, NY 11572, USA

**Keywords:** Myelodysplastic Syndrome, hematological malignancies, personalized medicine, next generation sequencing, cancer, precision oncology, immunotherapy

## Abstract

**Simple Summary:**

Current research is exploring the evolving landscape of Myelodysplastic Syndrome (MDS) treatment, a challenging condition often progressing to acute myeloid leukemia. For low-risk MDS, the focus is on personalized care through precise risk assessment and tailored interventions, utilizing means like erythropoiesis-stimulating agents, lenalidomide, and luspatercept. High-risk MDS treatments are shifting towards upfront doublet or triplet therapies and minimal residual disease (MRD) monitoring. A holistic approach integrates treatments like stem cell transplants and post-transplant maintenance, guided by individual patient circumstances. Precision medicine, driven by Next Generation Sequencing (NGS), aids in early diagnosis, prognosis, and customized interventions, allowing for investigations through clinical trials within more homogeneous patient cohorts characterized by similar molecular profiles. Based on these premises, the future of MDS treatment irrefutably moves towards personalized care, leveraging advanced technologies and molecular insights to enhance patient outcomes in the realm of hematological malignancies.

**Abstract:**

This perspective delves into the evolving landscape of Myelodysplastic Syndrome (MDS) treatment. MDS presents a significant clinical challenge, often progressing to acute myeloid leukemia. For low-risk MDS, the emphasis is on personalized care through comprehensive risk assessment, clinical monitoring, and tailored interventions, including promising agents like erythropoiesis-stimulating agents, lenalidomide, and luspatercept, with the anticipation of an expanding therapeutic arsenal and early intervention for improved outcomes. In contrast, high-risk MDS treatment is evolving towards upfront doublet or triplet therapies with a focus on minimal residual disease (MRD) monitoring. A holistic approach integrates various modalities, including stem cell transplant and post-transplant maintenance, all guided by individual patient circumstances. Risk-adapted strategies are crucial for enhancing patient outcomes. Precision medicine for MDS treatment is budding, largely driven by Next Generation Sequencing (NGS). NGS aids in early diagnosis, prognostication, and the targeting of specific mutations, with molecular data increasingly informing treatment responses and allowing for tailored interventions. Clinical trials within homogeneous patient groups with similar molecular profiles are becoming more common, enhancing treatment precision. In conclusion, the future of MDS treatment is moving towards personalized medicine, leveraging advanced technologies like NGS and molecular insights to improve outcomes in the realm of hematological malignancies.

## 1. Introduction

Myelodysplastic syndromes (MDSs) represent a diverse and complex group of neoplastic stem cell diseases that fall within the realm of hematological malignancies, commonly referred to as blood cancers. Characterized by bone marrow failure leading to cytopenias and associated complications, MDS is a critical clinical entity. Recent advancements in the field have led to the identification of clonality in nearly 90% of MDS patients, shedding light on the molecular underpinnings of this condition [1].

MDS holds a significant place in the landscape of myeloid malignancies and is estimated to affect approximately 50,000 individuals annually. Notably, MDS predominantly afflicts the elderly population, with an average age of onset around 70 years. Alarmingly, a substantial fraction of MDS patients, ranging from 30% to 40%, ultimately progress to acute myeloid leukemia, further accentuating the clinical relevance of this condition [2].

Tragically, MDS carries a considerable burden of morbidity and mortality, with up to 60% of patients succumbing to disease-related complications, often exacerbated by concurrent comorbidities. Historically, MDS faced challenges in gaining recognition as a form of cancer in official registries, although this situation has since been rectified. The nomenclature itself can be misleading, as “dysplasia” merely denotes morphological aberrations in cell appearance, rather than indicating a pre-cancerous state. Consequently, there is a growing belief that cases of MDS may go undiagnosed, particularly in the elderly population, where anemia, one of the hallmark features of MDS, ranks as one of the top five causes of this condition [1]. This research perspective is the synthesis of ideas and insights gained from an interview with Dr Rami Komrokji by the MedNews Week team focused on the future of MDS treatment with a stratified approach to improve treatment outcomes for patients.

## 2. Future of Low-Risk MDS Treatment

The future of low-risk MDS treatment is poised for a paradigm shift towards personalized care, driven by an enhanced understanding of clinical and molecular features. Comprehensive risk stratification, encompassing clinical assessment and molecular profiling, is becoming increasingly central to tailoring treatment strategies for lower-risk MDS patients. Emerging models like the IPSS-R and innovative approaches showcased at events such as the American Society of Hematology (ASH) annual meeting are offering refined tools to precisely gauge the risk profile of each patient [3].

In practice, the management of low-risk MDS involves a nuanced approach. Patients are closely monitored to assess the tempo of disease progression and to identify those with unfavorable outcomes [4]. This proactive stance allows for the consideration of higher-risk treatments in selected cases. Conversely, asymptomatic patients with stable blood counts do not warrant immediate intervention [3].

For patients exhibiting cytopenias, the choice of treatment hinges on the specific cytopenia type. Anemia is the predominant concern in the majority of cases, often leading to transfusion dependence over time. Erythropoiesis-stimulating agents are commonly employed, with lenalidomide reserved for those with a 5q deletion. Luspatercept specifically holds promise for patients with refractory anemia characterized by ring sideroblasts, as well as patients with spliceosome factor 3B1 (SF3B1) mutations. Exploring the potential of these agents in broader contexts, such as third-line therapy or in cases of concurrent thrombocytopenia and neutropenia, is underway [3].

Isolated neutropenia usually does not warrant immediate treatment unless recurrent infections become evident, in which case, limited options like thrombopoietin receptor agonists and cyclosporine (in younger patients) or hypomethylating agents (in older patients) may be considered [3].

In summary, the future of low-risk MDS treatment aligns with personalized approaches guided by precise risk assessment. Early intervention and preventive strategies concentrate on addressing precursor conditions like clonal hematopoiesis of indeterminate potential (CHIP) and clonal cytopenias of undetermined significance (CCUS). As new therapeutic agents continue to emerge, the therapeutic arsenal for lower-risk MDS patients is expected to expand, offering hope for improved outcomes and ultimately altering the trajectory of the disease.

## 3. Future of High-Risk MDS Treatment

To address the challenges of higher-risk MDS, it is essential to set higher standards and adopt a more comprehensive approach. We need to think beyond the current standards and consider upfront doublet or triplet therapy in treatment regimens [5,6]. Additionally, the growing ability to measure minimal residual disease (MRD) offers an exciting opportunity to customize treatments for each patient.

A total therapy approach should be the goal. This approach involves integrating various treatment modalities into a cohesive strategy. It begins with upfront treatment, which can include newer drugs and combinations. For some patients, an allogeneic stem cell transplant remains a vital component, especially if their MRD status and molecular profile indicate it is the best course of action [3].

Furthermore, the concept of maintenance therapy after transplant is under investigation. By closely monitoring MRD and risk factors, we can determine whether patients may benefit from post-transplant maintenance. MRD negativity might suggest that some patients could thrive without a transplant, and instead, they could continue with maintenance therapy.

On the other hand, data from acute lymphocytic leukemia suggests that patients achieving MRD negativity might benefit the most from a transplant. Therefore, a risk-adapted approach is crucial, tailoring treatments to the individual patient’s circumstances [7].

In this era of advanced molecular profiling, we have a wealth of information at our disposal to guide treatment decisions. Improvements in the utilization of this wealth of data could potentially lead to better measurements of MRD in higher-risk MDS. By leveraging this knowledge effectively, we can usher in a new era of personalized care for higher-risk MDS patients. It is about changing the treatment bar, thinking holistically, and using every tool in our arsenal to improve patient outcomes and quality of life.

## 4. Precision Medicine for MDS Treatment

In the current era, access to advanced technologies, particularly Next Generation Sequencing (NGS), has transformed the landscape of Myelodysplastic Syndrome (MDS) management. This trend is expected to further evolve, ushering in a new era of precision medicine.

Dr. Komrokji highlighted the integration of NGS into routine MDS patient evaluation. While NGS is not yet a formal diagnostic criterion, its role is rapidly expanding, especially in pre-MDS stages [3]. NGS provides an objective means of detecting clonal events, aiding in early diagnosis.

Furthermore, NGS plays a pivotal role in prognostication. It enables clinicians to comprehensively understand the disease spectrum within individual patients, identifying those with more favorable or adverse prognoses. These data empower the targeting of specific mutations, a practice already well established in acute myeloid leukemia (AML) and now extending to MDS. Table 1 demonstrates the spectrum of prognostication possible with NGS as well as descriptions of some of the variability in the disease manifestation from the different mutations.

Traditionally, MDS treatment approaches were relatively uniform, categorized into lower or higher risk groups. However, the advent of molecular profiling has unveiled a striking heterogeneity in clinical and molecular phenotypes among MDS patients. Consequently, there is a growing emphasis on conducting clinical trials within more homogeneous patient groups who share similar molecular characteristics. Table 2 outlines a list of mutations with specific responses to therapies that could be considered along the evolution of a patient’s disease course.

Moreover, molecular data are being incorporated into the assessment of treatment responses. Monitoring disease status using molecular markers, such as minimal residual disease, is becoming increasingly refined [3]. Patients achieving deeper molecular remissions tend to exhibit better outcomes, highlighting the value of molecular data in treatment response evaluation.

In summary, the trajectory of MDS management is unequivocally moving towards personalized medicine. The integration of molecular data into the diagnosis, risk stratification, and assessment of treatment responses promises to revolutionize the individualization of MDS therapies, offering patients more tailored and effective treatments.

## 5. Conclusions

In summary, MDS constitutes a significant challenge within hematological malignancies. The evolving approach to its treatment emphasizes tailoring interventions based on individual patients’ risk assessment and molecular profiling. This is of relevance in low-risk cases, while higher-risk MDS necessitates more comprehensive strategies, including innovative therapies and transplant options.

The incorporation of Next Generation Sequencing enhances precision medicine by enabling early diagnosis, accurate prognosis, and customized treatments. This growing reliance on molecular data to assess treatment responses holds promise for more personalized and effective therapeutic approaches.

Ultimately, the future of MDS management revolves around harnessing technology and molecular insights with the aim of enhancing patient outcomes.

## Figures and Tables

**Table 1 cancers-15-05170-t001:** Risk prognosis and disease associations of common mutations in MDS adapted from Bejar, R. 2014.

Single-Gene Mutations	Disease Association	Risk Prognosis
TP53	Complex and monosomal karyotype, excess blasts, thrombocytopenia, few mutations in other genes	Very Poor
SRSF2	More common in CMML	Poor
U2AF1	Often with del(20q)	Poor
DNMT3A		Poor
ASXL1		Poor
EZH2	More common in CMML	Poor
RUNX1	Thrombocytopenia, excess blasts	Poor
ETV6		Poor
NRAS/KRAS	Thrombocytopenia, excess blasts, monocytosis, more common in CMML, often subclonal	Poor
CBL	Monocytosis, excess blasts, more common in CMML	Poor
ZRSR2	On X chromosome, more common in males	Neutral
TET2	Normal karyotype, monocytosis, more frequent in CMML	Neutral
JAK2	50% of RARS-T, often subclonal	Neutral
SF3B1	Ring sideroblasts, fewer mutations in other genes	Good
IDH1/IDH2		Mixed
Cytogenetic Abnormalities	Disease Association	Risk Prognosis
>3 Abnormalities	Monosomal karyotype, TP53 mutation	Very Poor
3 Abnormalities		Poor
der(3q)	Often rearrangements near EV11-MDS1 locus	Poor
Deletion 7	Often part of monosomal karyotype	Poor
del(7q)	Often part of complex karyotype	Intermediate
Trisomy 8	Rare autoimmune or aplastic features, common in other myeloid malignancies	Intermediate
del(5q)	Isolated anemia, normal to elevated platelet count, but often part of complex karyotype	Good
del(20q)	Common in other myeloid malignancies	Good
del(11q)		Very Good
Deletion Y		Very Good

**Table 2 cancers-15-05170-t002:** Genes of interest for precision medicine in MDS and therapeutic considerations adapted from Garcia-Manero 2023.

Genes of Interest for Precision Medicine in MDS	
Genes	Decision-Making Considerations
SF3B1	Most commonly mutated gene in MDS. Luspatercept is indicated. New selective inhibitors of SF3B1 are being studied in clinical trials.
IDH1/IDH2	IDH inhibitors, Ivosidenib and Enasidenib indicated.
FLT-3	While rare for MDS, this mutation occurs in 15–30% of patients with HMA failure. Consider sorafenib with azacytidine in patients with HMA failure. More studies with FLT-3 inhibitors are in progress.
TP53	Major therapeutic need. Tends to be resistant to conventional chemotherapy with sensitive but short responses to HMA-based therapy. Prognosis is poor.
NPM1	Rare subset (~1% of MDS patients). Cytarabine followed by alloSCT should be considered.

## Data Availability

No patient data was directly utilized in this study.

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
