# Peer review of "The Evolving Landscape: Exploring the Future of Myelodysplastic Syndrome Treatment with Dr. Rami Komrokji"

_cancers, 2023, doi:10.3390/cancers15215170_

Round 1

Reviewer 1 Report

Comments and Suggestions for Authors

overall a very nice state of the field in MDS. The conclusion might be reviewed as some of the language is a little repetitive (e.g. personalized being used multiple times etc). Overall a good review.

Author Response

Thank you for your feedback. The aim of our work is to be a commentary geared toward summarizing the latest developments within the field. Dr. Komrokji used the IPSS-M and NGS data to set the framework for the discussion.

overall a very nice state of the field in MDS. The conclusion might be reviewed as some of the language is a little repetitive (e.g. personalized being used multiple times etc). Overall a good review.

Thank you kindly for your comment and suggestion. We have improved the conclusions while maintaining the core message of emphasizing the move toward individualized care in MDS treatment.

Reviewer 2 Report

Comments and Suggestions for Authors

This is a nicely summarized commentary on MDS current landscape and future perspectives. I have a few comments/ideas:

1. I believe that it would be helpful to include some information in specific types of mutations that now define not only the prognosis but even disease's entity (i.e. SF3B1 or TP53). 

2. Particularly, the authors describe that patients with RS are eligible to receive luspatercept but even the presence of SF3B1 mutation with lower RS% is enough to give eligibility for this agent. I would include this information.

3. The authors talk about MRD monitoring but I am not sure we are so close to talk about MRD in high-risk MDS. Perhaps in AML yes but not in MDS, thus I would clarify that much more works is required for this. 

Author Response

Thank you for your feedback. The aim of our work is to be a commentary geared toward summarizing the latest developments within the field. Dr. Komrokji used the IPSS-M and NGS data to set the framework for the discussion.

“This is a nicely summarized commentary on MDS current landscape and future perspectives. I have a few comments/ideas:

1. I believe that it would be helpful to include some information in specific types of mutations that now define not only the prognosis but even disease's entity (i.e. SF3B1 or TP53). 

Thank you for your helpful suggestions and direction.

We appreciated the suggestion of enriching our work with additional data. We have therefore included a table of mutations that both define prognosis as well as a table for genes that can now be effectively targeted with therapies.

2. Particularly, the authors describe that patients with RS are eligible to receive luspatercept but even the presence of SF3B1 mutation with lower RS% is enough to give eligibility for this agent. I would include this information.

We have included the suggested information.

3. The authors talk about MRD monitoring but I am not sure we are so close to talk about MRD in high-risk MDS. Perhaps in AML yes but not in MDS, thus I would clarify that much more works is required for this.”

We have followed the advice and added the clarification.

Reviewer 3 Report

Comments and Suggestions for Authors

In this commentary, Jackewicz et al. have presented the opinions of Dr. Rami Komrokji from MedNews Week; however, no clear novelty comes up from this commentary, nor novel therapeutic strategies in MDS. Moreover, using the IPSS-M, NGS data are now officially included within risk stratification systems.

Comments on the Quality of English Language

Extensive English editing

Author Response

“In this commentary, Jackewicz et al. have presented the opinions of Dr. Rami Komrokji from MedNews Week; however, no clear novelty comes up from this commentary, nor novel therapeutic strategies in MDS. Moreover, using the IPSS-M, NGS data are now officially included within risk stratification systems.”

Thank you for your feedback. This work aims to be a commentary to summarize the latest developments within the field. Dr. Komrokji used the IPSS-M and NGS data to set the framework for the discussion. We are sorry that our intent has probably been misunderstood.

Round 2

Reviewer 3 Report

Comments and Suggestions for Authors

None